# Defactinib in Combination with Mitotane Can Be an Effective Treatment in Human Adrenocortical Carcinoma

**DOI:** 10.3390/ijms26136539

**Published:** 2025-07-07

**Authors:** Henriett Butz, Lőrinc Pongor, Lilla Krokker, Borbála Szabó, Katalin Dezső, Titanilla Dankó, Anna Sebestyén, Dániel Sztankovics, József Tóvári, Sára Eszter Surguta, István Likó, Katalin Mészáros, Andrea Deák, Fanni Fekete, Ramóna Vida, László Báthory-Fülöp, Erika Tóth, Péter Igaz, Attila Patócs

**Affiliations:** 1Department of Molecular Genetics and the National Tumour Biology Laboratory, National Institute of Oncology, Comprehensive Cancer Centre, 7-9 Rath Gyorgy utca, H-1122 Budapest, Hungary; liko.istvan@oncol.hu; 2Department of Oncology Biobank, National Institute of Oncology, Comprehensive Cancer Centre, 7-9 Rath Gyorgy utca, H-1122 Budapest, Hungary; fekete.fanni@oncol.hu (F.F.); vidaramona97@gmail.com (R.V.); 3HUN-REN-OOI-TTK-HCEMM Oncogenomics Research Group, 7-9 Rath Gyorgy utca, H-1122 Budapest, Hungary; lorinc.pongor@hcemm.eu; 4Department of Laboratory Medicine, Semmelweis University, 4 Nagyvarad ter, H-1089 Budapest, Hungary; klilla0216@gmail.com (L.K.); szaboborbala92@gmail.com (B.S.); kati.balla@gmail.com (K.M.); 5Cancer Genomics and Epigenetics Core Group, Hungarian Centre of Excellence for Molecular Medicine, 9 Budapesti ut, H-6728 Szeged, Hungary; 6Department of Pathology and Experimental Cancer Research, Semmelweis University, 26. Ulloi ut, H-1085 Budapest, Hungary; dezso.katalin@semmelweis.hu (K.D.); tita.danko@gmail.com (T.D.); sebestyen.anna@semmelweis.hu (A.S.); sztankovics.daniel@gmail.com (D.S.); 7Department of Experimental Pharmacology and the National Tumour Biology Laboratory, National Institute of Oncology, Comprehensive Cancer Centre, 7-9 Rath Gyorgy utca, H-1122 Budapest, Hungary; tovari.jozsef@oncol.hu (J.T.); sara.surguta@gmail.com (S.E.S.); 8Artificial Transporter Research Group, Institute of Materials and Environmental Chemistry, Research Centre for Natural Sciences, 2 Magyar Tudosok korutja, H-1117 Budapest, Hungary; deak.andrea@ttk.hu; 9Department of Surgical and Molecular Pathology and the National Tumour Biology Laboratory, National Institute of Oncology, Comprehensive Cancer Centre, 7-9 Rath Gyorgy utca, H-1122 Budapest, Hungary; bathory.fulop.laszlo@oncol.hu (L.B.-F.); dr.toth.erika@oncol.hu (E.T.); 10Department of Endocrinology, Faculty of Medicine, Semmelweis University, 26. Ulloi ut, H-1085 Budapest, Hungary; igaz.peter@semmelweis.hu; 11Department of Internal Medicine and Oncology, Faculty of Medicine, Semmelweis University, 2 Koranyi Sandor, utca, H-1083 Budapest, Hungary

**Keywords:** adrenocortical cancer, defactinib, focal adhesion signalling, targeted therapy, 3D model, drug repurposing

## Abstract

Adrenocortical carcinoma (ACC) is an aggressive cancer with a poor prognosis. Mitotane, the only FDA-approved treatment for ACC, targets adrenocortical cells and reduces cortisol levels. Although it remains the cornerstone of systemic therapy, its overall impact on long-term outcomes is still a matter of ongoing clinical debate. Drug repurposing is a cost-effective way to identify new therapies, and defactinib, currently in clinical trials as part of combination therapies for various solid tumours, may enhance ACC treatment. We aimed to assess its efficacy in combination with mitotane. We tested the combination of mitotane and defactinib in H295R, SW13, and mitotane-sensitive and -resistant HAC15 cells, using functional assays, transcriptomic profiling, 2D and 3D cultures, bioprinted tissues, and xenografts. We assessed drug interactions with NMR and toxicity in vivo, as mitotane and defactinib have never been previously administered together. Genomic data from 228 human ACC and 158 normal adrenal samples were also analysed. Transcriptomic analysis revealed dysregulation of focal adhesion along with mitotane-related pathways. Focal adhesion kinase (FAK) signalling was enhanced in ACC compared to normal adrenal glands, with *PTK2* (encoding FAK) upregulated in 44% of tumour samples due to copy number alterations. High FAK signature scores correlated with worse survival outcomes. FAK inhibition by defactinib, both alone and in combination with mitotane, showed effective anti-tumour activity in vitro. No toxicity or drug—drug interactions were observed in vivo. Combination treatment significantly reduced tumour volume and the number of macrometastases compared to those in the mitotane and control groups, with defactinib-treated tumours showing increased necrosis in xenografts. Defactinib combined with conventionally used mitotane shows promise as a novel combination therapy for ACC and warrants further investigation.

## 1. Introduction

Adrenocortical carcinoma (ACC) is a rare malignancy, with an estimated incidence of 0.5–2 new cases per million people per year [1]. The prognosis of ACC is unfavourable: the median overall survival (OS) rate of patients with metastatic disease is 15 months [1,2]. Complete surgical resection is the recommended first-line treatment according to the European Society for Medical Oncology (ESMO) and the European Network for Study of Adrenal Tumours (ENSAT) guidelines [1]. More than half of ACC patients who have undergone complete tumour removal have a risk of relapse as high as 75–80%, often with metastases [1,2]. Therefore, adjuvant mitotane is also part of the standard treatment regimen [1] and is the only Food and Drug Administration-approved drug for ACC. Mitotane is an adrenolytic drug that directly targets adrenocortical cells and also reduces cortisol levels in the blood by inhibiting steroid biosynthetic enzymes [3]. The current guideline highlights that the clinical value of adjuvant mitotane treatment is somewhat controversial due to the lack of sufficiently large cohorts of treated and control patients in the available studies [1]. The clinical benefit of adjuvant use of cytotoxic chemotherapy in ACC is currently being explored in selected high-risk patients and clinical trials [1]. It is used in metastatic cases but its overall effectivity is poor; therefore, its indication may be restricted to selected patients with a very high risk of recurrence within clinical trials [1].

Targeted treatments (e.g., IGF1R inhibitors) have yielded mostly disappointing results to date but could be promising in some ACC subgroups [4]. While immunotherapy has emerged as a promising therapeutic approach, the heterogeneous expression of PD1, PD-L1, and CTLA-4 in a large series of ACC cases may explain the varied results of immunotherapies in advanced ACC [5,6].

Given the high failure rates, costs, and slow pace of drug development, repurposing de-risked compounds offers a cost-effective strategy to accelerate drug approval [7]. Defactinib, a focal adhesion kinase (FAK) inhibitor, is currently being investigated in several clinical trials, including multiple phase 1 and 2 trials for advanced solid tumours, i.e., KRAS-mutant non–small cell lung cancer and malignant mesothelioma [8,9]. It has been proven beneficial in high-grade ovarian and pancreatic cancer in phase 2 studies and is currently being tested in a phase 3 study (NCT06072781). Single-agent FAK inhibitors have shown limited efficacy, mainly because FAK functions as a signalling hub enabling cancer cell adaptation to treatment stress. However, combining FAK inhibitors with other therapies may be an effective therapeutic strategy [8]. Indeed, this approach is under investigation in combination regimens—for instance, with the RAF/MEK inhibitor avutometinib in recurrent low-grade serous ovarian cancer. Encouraging phase 1 data from this combination led to a breakthrough therapy designation and initiation of a phase 2 trial [10]. Moreover, defactinib is being studied in combination with immunotherapies, such as pembrolizumab, in patients with advanced solid malignancies [11].

Defactinib (also known as VS-6063 or PF-04554878) is an ATP-competitive inhibitor of FAK [9]. It functions by blocking FAK autophosphorylation—particularly at tyrosine residues Y397 and Y925—thereby disrupting integrin-mediated signalling cascades, including the RAS/MEK/ERK and PI3K/Akt pathways, which are involved in tumour cell proliferation, survival, migration, and angiogenesis [12]. Preclinical studies have demonstrated potent anti-tumour activity of defactinib both in vitro and in vivo [12].

The Wnt/β-catenin signalling pathway plays a central role in ACC tumorigenesis, with activating mutations frequently observed [13], yet identifying effective Wnt inhibitors for clinical use remains a challenge [14,15]. Inhibiting aberrant Wnt/β-catenin activity disrupts tumour microenvironmental reprogramming and reduces ACC growth [15]. The interaction between FAK and Wnt signalling is highly context-dependent: in intestinal tumorigenesis, FAK promotes Wnt activity, while in homeostasis, Wnt regulates FAK [14]. In different cancers, FAK can either modulate or act downstream of Wnt, and in mesothelioma, the two pathways are antagonistic [14]. These diverse interactions support the potential efficacy of FAK inhibition by defactinib in ACC [14].

As mitotane and defactinib have never been administered together in any tumour type, we conducted in vitro 2D and Matrigel-scaffolded 3D cultures, bioprinted tissues, and in vivo xenografts to assess their potential effectiveness, toxicity, and drug–drug interactions.

## 2. Results

### 2.1. Transcriptome Sequencing Identifies FAK Signalling in Mitotane-Treated In Vitro ACC Models

We generated in vitro 2D monolayer and 3D Matrigel-scaffolded models using the H295R adrenocortical cell line (Appendix A). Mitotane treatment was confirmed to reduce proliferation and hormone production in the 2D model, with increased dead cells (Appendix A–D). In the 3D Matrigel model, mitotane also suppressed cell proliferation and hormone secretion, though less effectively, with no significant effect on the dead-cell ratio (Appendix A–G).

Transcriptome sequencing of both models revealed gene enrichment in response to mitotane. Using commonly changed genes in both models (Figure 1A), cell adhesion, focal adhesion, and extracellular matrix (ECM)-associated processes were significantly altered along with mitotane effect-related steroid and lipid metabolism (Figure 1B,C, Table 1 and Appendix A).

To validate these findings, we examined ECM and FAK expression in silico using a mitotane-resistant HAC15 adrenocortical model generated through long-term mitotane treatment and clonal selection [16]. The data showed no changes in mitotane-sensitive cells, but the cell adhesion molecules pathway was significantly enriched in mitotane-resistant cells (Appendix A). In mitotane-treated H295R xenografts [17], only epithelial cell differentiation-related processes were altered compared to controls (Appendix A).

Based on these results, FAK signalling was selected for further analysis in human adrenocortical tumour samples to assess the potential effectiveness of defactinib, a FAK inhibitor.

### 2.2. Functional and Prognostic Relevance of Focal Adhesion Signalling in Human Adrenocortical Carcinoma Samples

We analysed 79 ACC samples and 128 normal adrenal gland samples to assess the relevance of FAK signalling (Figure 2A–C, Appendix A).

In ACC, 44% (39/89) of tumours showed FAK encoding *PTK2* gene amplification, leading to increased *PTK2* expression compared to diploid ACCs (*p* < 0.0001) and those with shallow deletions (*p* < 0.0001), (Figure 2A and Appendix A). DNA methylation analysis suggested that *PTK2* expression is mainly regulated by copy number variation (CNV) rather than methylation (R^2^ = 0.01675, *p* = 0.2588, Appendix A).

Gene enrichment analysis revealed pathways related to cell junctions, cell adhesion, focal adhesion, and ECM in ACC compared to normal adrenal glands (Appendix A). FAK-related genes, including TGFβ, TP53 signalling, and gene transcription factors, were significantly correlated with *PTK2* expression (Appendix A). Distinct profiles of cell adhesion and motility were observed in *PTK2*-high and *PTK2*-low ACC samples (Figure 2B, Appendix A). Moreover, genes regulating adhesion, ECM, Wnt signalling, and migration were differentially methylated in *PTK2*-high versus *PTK2*-low tumours (Figure 2C, Appendix A).

While *PTK2* expression alone was not a prognostic marker, high expression of other FAK signalling components (e.g., *COL1A*, *ITGA1*, *CTNNB1*) was linked to poorer survival (Appendix A). A FAK signature comprising the ssGSEA activity of 11 genes could distinguish normal and ACC samples and was associated with worse overall and progression-free survival (Figure 2D,E), even after adjusting for clinical characteristics using multivariate analysis (Appendix A).

These findings prompted further investigation of defactinib in combination with mitotane.

### 2.3. Defactinib Combined with Mitotane Shows Efficacy in ACC In Vitro Models

As defactinib has not been tested on ACC cell lines before, we performed concentration–response viability assays in adrenocortical carcinoma cell lines using 0.1, 0.5, 1, 2.5, and 5 µM defactinib (Appendix A). Based on our results, we continued to use 1 and 5 µM as effective doses. The combination of defactinib and mitotane inhibited H295R cell proliferation more effectively than either treatment alone, increasing the ratio of dead cells in both 2D and 3D culture models (Figure 3A,B). While no notable morphological changes were observed in monolayer cultures, in Matrigel-scaffolded spheroids the combination treatment reduced spheroid size compared to those in control and mitotane-only groups, with marked disruption of spheroid structure and reduced compactness, suggesting compromised cellular adhesion (Figure 3C–E and Appendix A). No significant difference in cortisol production was detected between mitotane-only and combination treatment groups (Appendix A). Both defactinib alone and the combination treatment inhibited cell migration (Figure 3F and Appendix A).

To validate these results, we tested the combination treatment on SW13 cells, observing similar outcomes (Figure 4). In SW13, while the combination did not outperform defactinib alone in 2D and 3D bioprinted models, it was more effective than both the control and mitotane-only treatments (Figure 4A,B). The increased efficacy of the combination in the 3D Matrigel-scaffolded model (Figure 4C) likely reflects differences in drug availability, as nutrients reach inner cells by diffusion in Matrigel-embedded spheroids, unlike in the bioprinted model. Similar to the case with H295R cells, the combination treatment in SW13 significantly reduced cell migration compared to those in the mitotane-alone and control groups (Figure 4D).

### 2.4. Defactinib Combined with Mitotane Is Effective in ACC Xenograft Model

No prior data on the mitotane-defactinib combination exist, so we conducted a drug–drug interaction and toxicity assessment. ^1^H NMR analysis showed no chemical reaction between defactinib and mitotane in DMSO-d6 solution, confirming that both drugs retained their structural integrity when mixed in corn oil using per os administration (Appendix A). A preliminary animal study with 3-3 animals receiving either the combination or corn oil, respectively, for two weeks showed no signs of toxicity.

In the H295R xenograft model, treatment with mitotane, defactinib, or the combination was administered for six weeks after tumour formation (Figure 5A). The combination of defactinib and mitotane significantly reduced tumour volume and the number of macrometastases compared to those in the mitotane treatment and control groups (Figure 5B–E and Appendix A). Tumours treated with defactinib also showed more extensive necrotic areas than those treated with mitotane or the control, indicating defactinib’s effect (Figure 5F,G). No significant differences were found in Ki67 indices between the treatment groups (Appendix A).

## 3. Discussion

Given the limited effectiveness of current ACC treatments, we explored the effect of a clinically potentially effective combined therapy of two drugs already available on the market. In vitro models (2D and Matrigel-scaffolded 3D) showed dysregulated cell adhesion and focal adhesion upon mitotane treatment which were also identified by in silico data from mitotane-sensitive and -resistant HAC15 cells.

FAK plays a pivotal role in cell migration, regulates cell adhesion and proliferation through integrin-–ECM interactions, and activates pathways like Wnt, Rho, PI3K/Akt, Ras/Raf/MAPK, and Hippo [8]. FAK also functions as an adaptor at focal adhesions and a nuclear scaffold in transcriptional regulation, including stress responses involving p53 [8]. Additionally, FAK influences chemokine expression, affecting the tumour microenvironment and anti-tumour immunity [8], which is of interest in ACC [5,6]. FAK also interacts with Wnt signalling, a key driver of tumorigenesis in ACC, with evidence suggesting bidirectional crosstalk between the two pathways [14].

Our analysis revealed that 44% of ACC cases exhibit amplification of the *PTK2* gene encoding FAK, leading to increased *PTK2* expression. *PTK2*-high- and *PTK2*-low-expressing ACC samples showed distinct profiles in adhesion-related pathways, though Wnt signalling differences were absent, likely due to mutations driving constitutive Wnt activation. Differential methylation analysis also indicated regulatory interactions between FAK and Wnt signalling that warrant further investigation [14].

While *PTK2* expression alone did not serve as a prognostic marker, tumours with high FAK activity were associated with worse progression-free, disease-specific, and overall survival [18,19]. These findings suggest that FAK activation, rather than *PTK2* expression alone, impacts patient outcomes. We showed that 88% of ACC samples either stably express (diploid) or overexpress (chromosomal amplification or gain) *PTK2*. *PTK2* expression remained detectable even in samples with shallow deletions. FAK inhibition could therefore be beneficial in ACC.

As defactinib has not been previously tested on ACC cell lines, to establish appropriate treatment concentrations, we reviewed the literature on defactinib use across cancer models, where in vitro studies commonly applied concentrations ranging from 0.1 to 20 µM [20,21,22,23,24]. Based on this, we performed concentration–response viability assays in adrenocortical carcinoma cell lines, and finally, concentrations of 1 and 5 µM defactinib were used as effective doses. For mitotane, we used 5 µM based on our previous studies and published literature demonstrating its effectiveness in modulating cell viability and transcriptomic responses in vitro [25,26,27], which concentration has also been widely applied in 3D cell culture models [28]. The defactinib–mitotane combination effectively inhibited cell proliferation and cell migration. Slight differences in response may be due to cell line origins (H295R being hormone-producing from a primary tumour and SW13 from a metastasis). Nevertheless, xenograft studies confirmed the efficacy of the combined therapy, significantly reducing tumour volume and number of macrometastases and increasing tumour necrosis. Altogether our findings indicate correlations across different experimental models regarding the beneficial effect of the combined therapy versus mitotane-only conditions. Specifically, combined treatment led to reduced cell migration in 2D monolayer cultures, impaired adhesion in 3D ECM-embedded spheroids, and decreased tumor size and number of macrometastases in the xenograft model. These are consistent with transcriptomic data from human ACC samples, where high FAK expression was associated with poorer progression-free and overall survival—supporting the role of FAK in promoting tumour progression through enhanced cell migration and metastatic potential.

Mitotane and defactinib exert their effects through distinct molecular pathways—mitotane primarily inhibits steroid biosynthesis, while defactinib targets cell migration via FAK inhibition. Therefore, our findings suggest that mitotane’s adrenolytic and steroid inhibitor effects can be complemented by defactinib, which inhibits cell adhesion, proliferation, migration, and metastasis development. In literature, in line with our findings, a combinatorial approach to target FAK is a frequently used strategy in clinics [8]. Indeed, in lung, ovarian, prostate, and pancreatic cancers, FAK inhibitors combined with other anti-cancer drugs significantly reduced tumour growth, tumour volume, and tumour load; increased apoptosis; and improved survival in in vitro and in vivo studies [23,29,30,31].

Defactinib is now being investigated in several phase 1 and phase 2 studies [9], and currently in a phase 3 trial (NCT06072781) for ovarian cancer, it has shown promise in overcoming treatment resistance and has received orphan drug and breakthrough designations from the FDA [8,32,33,34,35]. FAK inhibitors, which are well tolerated and orally administered [36,37], have not been studied in combination with mitotane previously. We confirmed no chemical interactions or degradation between the two drugs, and no toxicity was observed in vivo. Recent studies suggest that FAK inhibition could enhance tumour cell sensitivity to immune responses, potentially through a FAK–PD-L1 mechanism [38]. Therefore, FAK inhibition may offer broader therapeutic potential, warranting further studies on its combination with chemotherapy or immunotherapy in ACC.

Our study has several limitations. We used different adrenocortical cancer cell lines: H295R, which is the most accepted ACC model exhibiting steroidogenic activity, and the SW13 cell line, which does not exhibit steroidogenic activity but has continued to serve as a valid and used model for ACC, particularly in pharmacological studies. Recent literature (2022–2025) [39,40,41,42,43,44,45,46,47,48] supports its relevance, with several peer-reviewed publications employing SW13 alone or in combination with H295R to explore ACC pathophysiology and treatment response. Moreover, the absence of hormone production in SW13 reflects a subset of human ACCs, thereby contributing to the representation of tumour heterogeneity. In our study, both SW13 and H295R responded to mitotane–defactinib combination therapy, highlighting the potential therapeutic value across biologically distinct ACC models. In our in vitro analysis we included a comprehensive series of experiments: two independent xenograft models, 2D monolayer cultures, 3D spheroids in ECM, 3D bioprinted tissue models, and transcriptomic analyses of both mitotane-sensitive and -resistant HAC15 cells supplemented by human ACC tissue data that provide robustness to the multi-level data. Nevertheless, we acknowledge the limitations of individual cell lines and explicitly note the importance of future validation in additional, well-characterised models.

Another limitation is related to the potential pharmacokinetic interactions. Mitotane is a known strong inducer of hepatic CYP3A4, while defactinib is metabolized via CYP3A4 and CYP2C9 [49]. Therefore, it is plausible that mitotane could accelerate the metabolism of defactinib, potentially reducing its plasma levels and therapeutic efficacy. Shimizu et al. in a defactinib phase 1 study highlighted that preclinical evidence indicated that VS-6063 has a low potential for CYP3A-mediated drug interactions; in their study, potent CYP3A4 inhibitors or inducers and systemic anticoagulation were prohibited [37]. In our study, no direct drug–drug interaction was observed in vitro using ^1^H NMR spectroscopy, and our in vivo toxicity study of the combined treatment for three weeks did not show clinical evidence of toxicity in mice. Nevertheless, in our in vivo model, we still observed a significant benefit of the combination therapy over mitotane monotherapy in terms of both tumour volume and metastatic burden. Still, the possibility of a pharmacokinetic interaction in vivo in human underscores the importance of evaluating drug metabolism and interactions in future preclinical/clinical studies. While our current work focused on assessing therapeutic efficacy, we fully recognize the need for detailed pharmacokinetic investigations to better understand the impact of CYP3A4 induction on defactinib when co-administered with mitotane.

Despite strong preclinical evidence, human clinical validation and pharmacokinetic studies are needed. Given ACC’s rarity, clinical trials will require collaborative efforts. Further preclinical research should explore defactinib’s potential with immunotherapy or chemotherapy. Drug repurposing remains an effective strategy, offering reduced development time and costs, especially for rare cancers in need of new treatments [7].

## 4. Materials and Methods

### 4.1. In Vitro 2D and 3D ACC Models, Bioprinting, Treatments and Functional Assays

H295R (RRID: CVCL_0458) and SW13 (RRID: CVCL_0542) adrenocortical cell lines (initiated from female patients) were obtained from LGC Standards GmbH (Wesel, Germany). Both cell lines are accepted and valid models for ACC-related research [39,40,41,42,43,44,45,46,47,48]. Cell culturing, Matrigel scaffolding, and 3D bioprinting were done exactly as previously [50,51,52] and are described in detail in Appendix A. Imaging was done by Canon Power Shot A590 IS software using a ×50 objective and ×10 ocular.

Mitotane (SML1885, Merck-Sigma-Aldrich, Darmstadt, Germany) at 5 and 15 μM and defactinib (ab254452, Abcam, Cambridge, UK) at 1 μM and 5 μM diluted in dimethyl sulfoxide (DMSO, D4540, Merck-Sigma-Aldrich, Darmstadt, Germany) at a final 0.01 V/V% were used. As defactinib has not been previously tested on ACC cell lines, to establish appropriate treatment concentrations, we reviewed the literature on defactinib use across cancer models, where in vitro studies commonly applied concentrations ranging from 0.1 to 20 µM [20,21,22,23,24]. Based on this, we performed concentration–response viability assays in adrenocortical carcinoma cell lines (Appendix A), and finally, concentrations of 1 and 5 µM defactinib were used as effective doses.

For determining cell viability and proliferation, conventional Alamar blue assay (DAL1025, Invitrogen, Thermo Fisher Scientific, Grand Island, NY, USA), s/ulforhodamine B (SRB) assays, and trypan blue assays were used as previously [50,51] and as given in Appendix A. All measurements were done at least three times (biological replicates) with one to three technical replicates in each.

### 4.2. Xenograft Model and Dose Testing with Non-Lethal Outcomes

All procedures can be found in Appendix A. Briefly, the H295R xenograft model was established in SCID mice as previously (Supplementary Methods) [17]. After tumour formation, in the first experiment, 200 mg/kg/day mitotane treatment (*n* = 8) or corn oil (as vehicle control; *n* = 8) was administered per os (gavage) in a final volume of 80 μL for 5 weeks.

Prior to the combined treatment, a three-week in vivo dose-testing toxicology study was performed: 200 mg/kg/day mitotane in 40 μL corn oil was mixed with 50 mg/kg/day defactinib in 40 μL corn oil with 5% DMSO and administered to the animals by per os gavage. To investigate toxic effects and the tolerance of the combination therapy, we followed the guideline of Laboratory Animal Management and Welfare [53].

In the second xenograft experiment, treatment groups were designed as follows: (i) 200 mg/kg/day mitotane in 80 μL corn oil (*n* = 8), (ii) 50 mg/kg/day defactinib in 80 μL corn oil with 5% DMSO (*n* = 9), (iii) 200 mg/kg/day mitotane in 40 μL corn oil with 50 mg/kg/day defactinib in 40 μL corn oil with 5% DMSO (*n* = 9), and (iv) 80 μL corn oil (*n* = 9) control for 6 weeks.

In both experiments, tumour sizes were measured by calliper and tumour volume was calculated as width2 × length × 0.5.

In vivo experiments and dose testing with non-lethal outcomes were authorised by the National Council for Scientific Ethics in Animal Experiments PE/EA/801-7/2020, PEI/001/1738-3/2015, and PE/EA/1461-7/2020.

### 4.3. Steroid Hormone Measurements and Nuclear Magnetic Resonance (NMR) Spectroscopy

Steroid hormone measurements were done according to our previously published protocols [50,54]. A detailed description of the HPLC-MS/MS and ^1^H nuclear magnetic resonance (NMR) spectroscopy for testing drugs’ chemical interaction can be found in Appendix A.

### 4.4. Transcriptome Sequencing and Bioinformatics

Total RNA was extracted with a Qiagen MiRNeasy Mini kit (217004, Qiagen, Hilden, Germany).

Library preparation and sequencing were done as previously described [55]. Sequencing was run on the Illumina Novaseq platform (Illumina Inc., San Diego, CA, USA; NovaSeq 6000 SP 300 cycles (2 × 150 bp)). Bioinformatic analysis was done as previously [56] and detailed in Appendix A. Raw data will be uploaded to NCBI upon manuscript acceptance.

### 4.5. In Silico Datasets, Genomic Characterisation, and Statistical Analysis

For the assessment of gene expression alteration in normal adrenal gland, ACC, mitotane response, and resistance in silico datasets were obtained from NCBI Gene Expression Omnibus (Appendix A). For genomic characterisation (copy number analysis, DNA methylome, gene and protein array expression profile analysis), genomic data of 92 human adrenocortical cancer samples were retrieved from The Cancer Genome Atlas through cBio Portal (https://www.cbioportal.org/; accessed on 25 July 2024). Detailed descriptions of used bioinformatical algorithms [57,58] and statistical analysis can be found in Appendix A.

## 5. Conclusions

Our preclinical results, along with human ACC genomic and survival analyses, suggest that FAK inhibition with defactinib can be effective in treatment of ACC. Defactinib showed no interaction with mitotane and potentiated its effect; therefore, combined therapy could also be a valid approach. These findings provide a foundation for further investigation of defactinib in combination with other cytotoxic therapy or immunotherapy for ACC similarly to ovarian and pancreatic cancers, where defactinib currently in phase 2–3 trials.

## Figures and Tables

**Figure 1 ijms-26-06539-f001:**
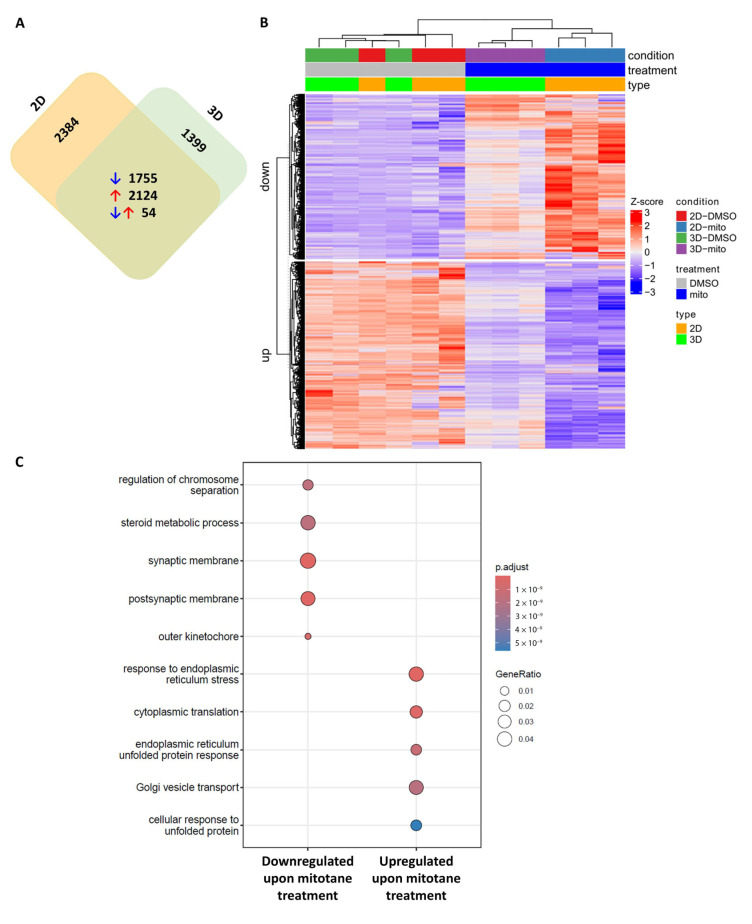
Transcriptome analysis of mitotane’s effects in 2D and Matrigel-scaffolded 3D models of H295R adrenocortical cells. (**A**) Gene expression changes following mitotane treatment in in vitro 2D cultures and Matrigel-scaffolded 3D ACC spheroids. A total of 1755 genes were commonly downregulated (blue arrows), and 2124 were upregulated (red arrows), while 54 genes exhibited opposing expression patterns (red & blue arrows) between the two models. (**B**) Heatmap of commonly altered gene expression in 2D and 3D cultures. (**C**) Most significant biological processes commonly affected by mitotane treatment in 2D and 3D models.

**Figure 2 ijms-26-06539-f002:**
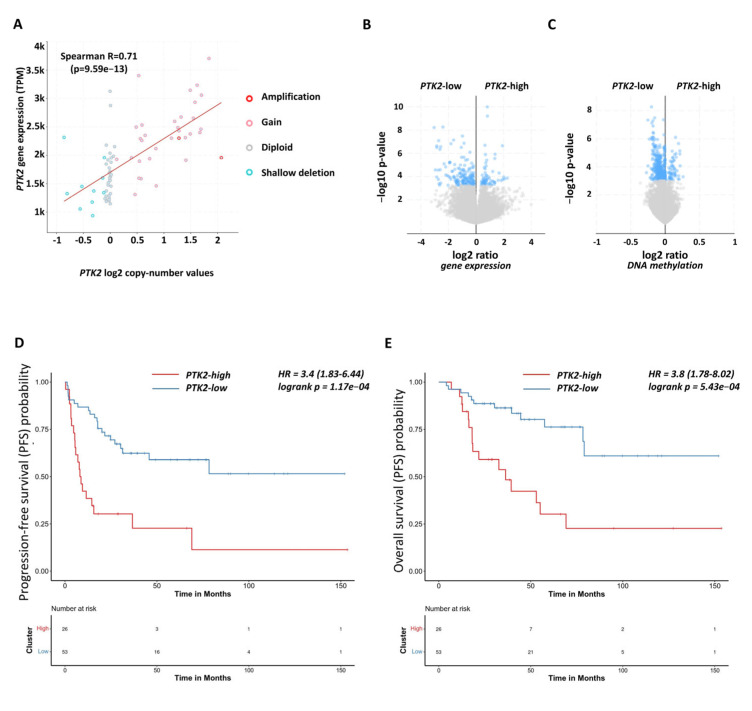
Expression of focal adhesion kinase (FAK)-encoding *PTK2* in human ACC. (**A**) Correlation between PTK2 expression and copy number alteration indicated a minor epigenetic effect in *PTK2* gene expression regulation. Differentially expressed genes (**B**) and differentially methylated genes (**C**) between *PTK2*-high- and *PTK2*-low-expressing tumours (*PTK2*-high samples were selected with chromosomal gain or amplification of the *PTK2* encoding region; *PTK2*-low samples were selected with diploid and shallow deletion of the *PTK2* encoding region—see Appendix A). The exact gene lists and their function are presented in Appendix A–F. Blue dots indicate significant, while grey dots indicate non-significant changes (**D**,**E**) The effect of FAK signature on ACC patient survival. FAK signature represents FAK activity characterised with 11 genes using ssGSEA. Patients with high FAK signature have worse progression-fee (**D**) and overall survival (**E**) in univariate and multivariate analyses (Appendix A), respectively.

**Figure 3 ijms-26-06539-f003:**
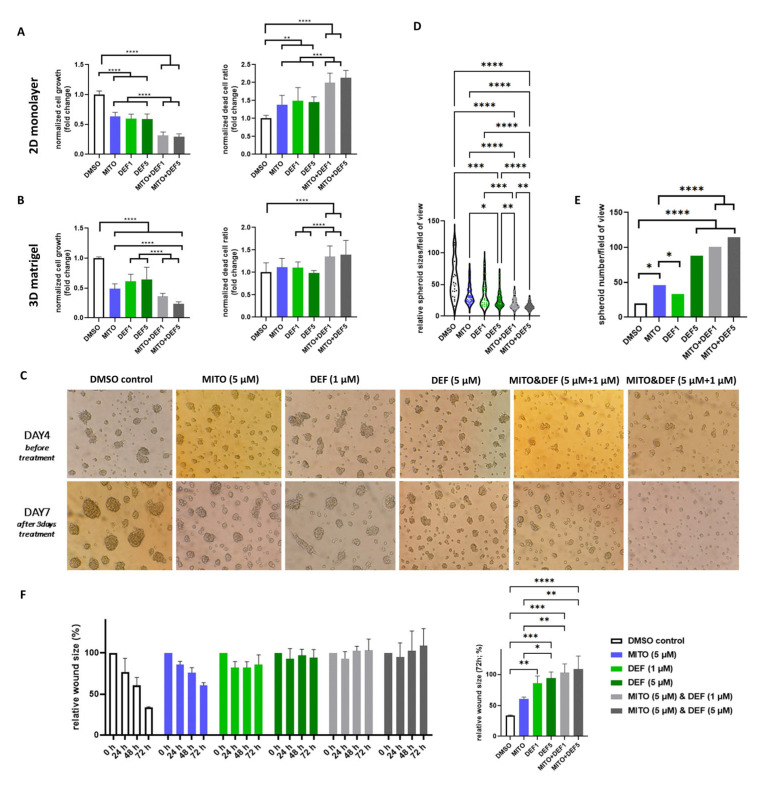
In vitro functional assessment of defactinib, mitotane, and the combined effect on H295R human ACC cell line. Cell proliferation and dead-cell ratio upon treatments in (A) 2D model and in (**B**) 3D Matrigel-scaffolded model. (**C**) Photomicrographs of Matrigel-scaffolded 3D models indicated smaller spheroid sizes (**D**) and increased spheroid numbers (**E**) upon combined treatment. Imaging was done by Canon Power Shot A590 IS v. 1.1.0.0 software using a ×50 objective and ×10 ocular. (**F**) Defactinib, mitotane, and combined treatment effect on cell migration. *: *p* = 0.01–0.05; **: *p* = 0.001–0.01; ***: *p* = 0.0001–0.001; ****: *p* < 0.0001

**Figure 4 ijms-26-06539-f004:**
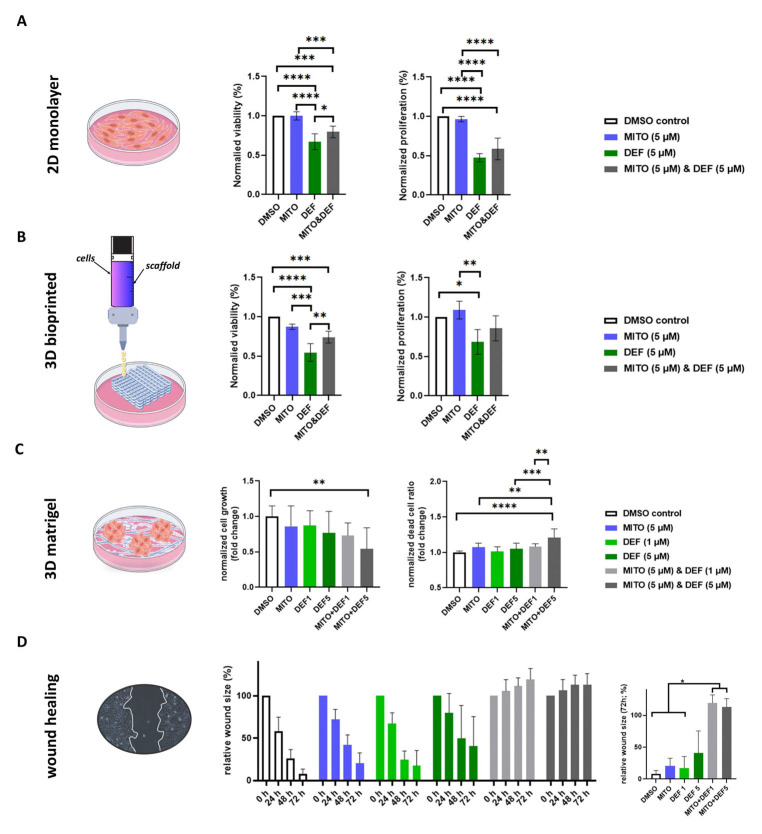
In vitro functional assessment of defactinib, mitotane and combined effect on SW13 ACC cell line. Treatment effect on viability and proliferation in 2D monolayer cultures (**A**) and in 3D bioprinted tissues (**B**). Proliferation and dead-cell ratio in 3D Matrigel-scaffolded models (**C**). (**D**) Defactinib, mitotane, and combined treatment effect on cell migration in time lapse (left side) and at the endpoint among different groups (right side). *: *p* = 0.01–0.05; **: *p* = 0.001–0.01; ***: *p* = 0.0001–0.001; ****: *p* < 0.0001.

**Figure 5 ijms-26-06539-f005:**
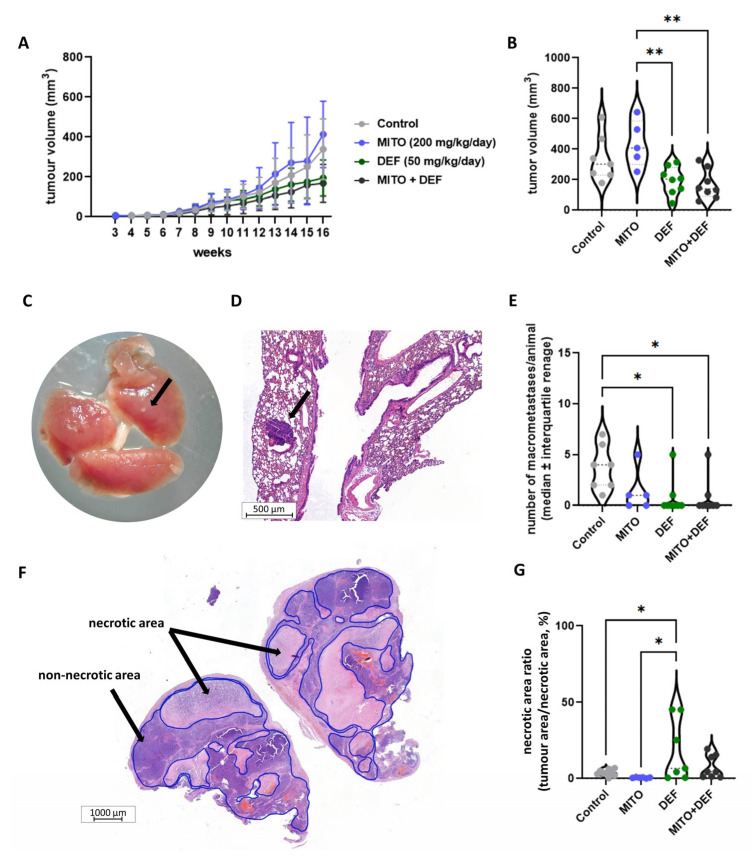
Defactinib, mitotane, and combined treatment effect on H295R xenografts. Tumour volume during the experiment (**A**) and at termination (**B**). Metastases were developed during the 16-week experiments. (**C**) Black arrow indicates a marcometastasis in the lung. Metastases were observed during serial sectioning, indicated by the black arrow (**D**). (**E**) The number of macrometastases was decreased in the combined-treatment groups compared to controls, while mitotane itself had not such effect. (**F**) Necrotic area of the tumours was identified by histology and quantified by image analysis. (**G**) Defactinib increased the necrotic area compared to mitotane only and control group. *: *p* = 0.01–0.05; **: *p* = 0.001–0.01

**Table 1 ijms-26-06539-t001:** Cell and focal adhesion-related gene ontology terms based on shared gene expression changes in 2D and 3D models following mitotane treatment. Full gene ontology and gene set enrichment results can be found in Appendix A.

ID	Ontology	Description	*p*.Adjusted	*q* Value
GO:0007267	BP	cell–cell signalling	0.00025	0.028
GO:0031012	CC	extracellular matrix	0.00087	0.058
GO:0098742	BP	cell–cell adhesion via plasma membrane adhesion molecules	0.00137	0.078
GO:0098640	MF	integrin binding involved in cell–matrix adhesion	0.003010	0.129
GO:0007155	BP	cell adhesion	0.01168	0.251
GO:0062023	CC	collagen-containing extracellular matrix	0.01543	0.286
GO:0099560	BP	synaptic membrane adhesion	0.01856	0.311
GO:0086019	BP	cell–cell signalling involved in cardiac conduction	0.02338	0.343
GO:0007156	BP	homophilic cell adhesion via plasma membrane adhesion molecules	0.02461	0.349
GO:0098609	BP	cell–cell adhesion	0.02524	0.356
GO:0030020	MF	extracellular matrix structural constituent conferring tensile strength	0.03141	0.385
GO:0030198	BP	extracellular matrix organization	0.03805	0.415
GO:0005925	CC	focal adhesion	0.04605	0.455

BP: biological process, MF: molecular function, CC: cellular component.

## Data Availability

All data are available in the main text or the Appendix A. Transcriptome sequencing data will be deposited in NCBI GEO upon acceptance.

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
