# Peer review of "Defactinib in Combination with Mitotane Can Be an Effective Treatment in Human Adrenocortical Carcinoma"

_ijms, 2025, doi:10.3390/ijms26136539_

Round 1

Reviewer 1 Report

Comments and Suggestions for Authors

Butz et al investigated the combination of mitotane and defactinib for the treatment of ACC.  The article is very well written and this drug combination is according to their results a new, promising treatment option. 

They presented conclusive cell culture data and also transcriptome data which show how effective this combination can be. 

The Authors described the methods in detail. Their methods are very well written. I would like to know why they choose the H 295 R cell line ?

It is also very interesting that they created a mitotane resistant cell line.

Maybe I missed it, but were the concentrations of mitotane and defactinib comparable to concentrations which could be administered in animal studies?

Furthermore, the authors show some preliminary results of studies already using Defactinib in other cancer types.
This shows that it is a tolerable and maybe effective drug.
The authors conclude that a combination with mitotane and defactinib should be suggested. I agree with this regarding their results of the cell culture experiments.

I have no further comments and agree with their findings and conclusion based on their results. 

Author Response

REVIEWER 1

First, we would like to thank the Reviewer for his/her accurate work during the review process.

Butz et al investigated the combination of mitotane and defactinib for the treatment of ACC.  The article is very well written and this drug combination is according to their results a new, promising treatment option.

They presented conclusive cell culture data and also transcriptome data which show how effective this combination can be.

The Authors described the methods in detail. Their methods are very well written.

Response: We are grateful the Reviewer for his overall positive opinion.

I would like to know why they choose the H 295 R cell line ?

Response: We selected NCI H295R cell line as i) they are among the most frequently used adrenocortical cell line (in NCBI Pubmed for query H295 AND ACC there is 55 scientific research paper published between 2020-2025), ii) it is commercially available from ATCC which is a reliable source for cell line purchase, and iii) in line with the most clinical cases it retains the ability to produce adrenocortical hormones cortisol and androgens.

It is also very interesting that they created a mitotane resistant cell line. Maybe I missed it, but were the concentrations of mitotane and defactinib comparable to concentrations which could be administered in animal studies?

Response: As mitotane-resistant cell lines were previously generated by Seidel et al. (2020, Endocrine Connections, DOI: 10.1530/EC-19-0510), we did not replicate the experiment. Instead, we utilized the publicly available transcriptomic data of mitotane-sensitive and -resistant HAC15 adrenocortical cells, which can be found in Supplementary Table 1. This way an additional cell line is also strengthen our findings using H295R and SW13 cell lines.

In the study of Seidel et al., a long-term, pulsed in vitro treatment with 70 µM mitotane was applied, representing approximately 1.5 times the IC₅₀ value. This concentration is slightly higher than the physiological target plasma concentration in humans, which is approximately 50 µM (equivalent to 14 mg/L), as reported by Hescot et al. (2014, DOI: 10.1530/ERC-12-0368). In in vivo H295R xenografted mouse experiments, Doghman et al. (2013, Molecular and Cellular Endocrinology, DOI: 10.1016/j.mce.2013.07.023) demonstrated that this target plasma concentration (50 µM) can be achieved through daily oral administration of 200 mg/kg mitotane – the same dose we used in our xenograft study. This dosing regimen is consistent with what is commonly applied in similar xenograft mouse studies reported in the literature.

As a conclusion the dose used to generate mitotane-resistant cells are slightly higher compared to the physiologically achieved concentration, although IC₅₀=1.5 is still is on the lower end, as developing drug-resistant cell lines, researchers typically aim for a ≥ 3-fold increase in IC₅₀ compared to the parental line to ensure stable and meaningful resistance according to general recommendation (McDermott et al. 2014; DOI: 10.3389/fonc.2014.00040).

Furthermore, the authors show some preliminary results of studies already using Defactinib in other cancer types.

This shows that it is a tolerable and maybe effective drug. The authors conclude that a combination with mitotane and defactinib should be suggested. I agree with this regarding their results of the cell culture experiments.

I have no further comments and agree with their findings and conclusion based on their results.

Response: We greatly appreciate Reviewer 1’s opinion and positive feedback on our work. We are pleased that the Reviewer agrees with our conclusions regarding the potential combination of mitotane and defactinib, as supported by our preclinical study.

Reviewer 2 Report

Comments and Suggestions for Authors

In the manuscript titled “Defactinib in combination with mitotane can be an effective treatment in human adrenocortical carcinoma” the authors investigate the effect of defactinib in combination with mitotane in ACC models.

Despite an interesting transcriptome sequencing analysis that justify the rational to test defactinib in the ACC contest, some parts require substantial revision and improvement.

Major points:

  1. SW13 is no longer considered a valid ACC model. Several new ACC cell lines are now available to the scientific community upon request from the respective owners and can be obtained easily and free of charge (DOI: 10.1007/s12020-022-03112-w). Unfortunately, I cannot accept a manuscript that includes only two ACC cell models, one of which is SW13. The authors must replace SW13 with another, more appropriate ACC cell model. By the way the results are not in line with those obtained in H295R.
  2. Regarding the concentrations of mitotane and defactinib used in the 2D and 3D experiments, the authors must explain how these concentrations were selected. Were preliminary experiments conducted using increasing concentrations of the drugs, particularly defactinib? I recommend including the concentration–response curves, at least those obtained in monolayer cultures.
  3. In the in vivo experiments, there appear to be no significant differences between the group treated with defactinib alone and the group treated with the combination with mitotane. How do your results support the rationale for combining these treatments? Furthermore, from a clinical perspective, defactinib is metabolized by hepatic CYP3A4. Although mitotane is also primarily metabolized and cleared by the liver, one of its most notable effects is the induction of CYP3A4 activity. Therefore, mitotane could significantly influence drug metabolism and potential drug–drug interactions.

Minor points:

  1. In the abstract (line 29), the phrase “but its impact on outcomes is limited” in reference to mitotane should be reconsidered. I understand that the authors aim to convey that patient outcomes in ACC remain poor despite standard treatments. However, mitotane has been shown to provide clinically significant benefits, particularly when surgery is not feasible or when used as adjuvant therapy to prevent or delay disease recurrence. Mitotane remains the cornerstone of systemic treatment for ACC, both in the unresectable/metastatic setting and as adjuvant therapy.
  2. In the introduction (line 66), the phrase “The use of cytotoxic drugs in ACC is not well-established [1]” is not clear to me. What do the authors mean by this sentence?
  3. Please improve the section of the introduction related to defactinib mechanism of action and its involvement in clinical trials. Moreover, consider adding a sentence that connects its mechanism of action to the Wnt/β-catenin signaling pathway, preferably between lines 82 and 83. Alternatively, you may rephrase the sentences following line 83 to better establish this link.
  4. In the results section (lines 168-169) the authors affirm “the combination treatment reduced spheroid size compared to control and mitotane-only groups, with an increased number of spheroids suggesting potential adhesion inhibition”. The increased number of spheroids should be reconsidered and replaced with something like “marked disruption of spheroid, reduction in compactness indicative of compromised cellular adesion”.
  5. The Discussion section could be expanded and further developed. Additionally, the sentence in line 254 is identical to the one in the Introduction (line 81); consider rephrasing or removing it to avoid repetition.

Author Response

REVIEWER 2

First, we would like to thank the Reviewer for his/her accurate work during the review process.

In the manuscript titled “Defactinib in combination with mitotane can be an effective treatment in human adrenocortical carcinoma” the authors investigate the effect of defactinib in combination with mitotane in ACC models.

Despite an interesting transcriptome sequencing analysis that justify the rational to test defactinib in the ACC contest, some parts require substantial revision and improvement.

Response: We thank the Reviewer for his/her valuable feedback and for recognizing the relevance of our transcriptomic analysis. We appreciate the constructive comments and carefully have revised the manuscript to address all the points raised to improve the overall quality of the work.

Major points:

SW13 is no longer considered a valid ACC model. Several new ACC cell lines are now available to the scientific community upon request from the respective owners and can be obtained easily and free of charge (DOI: 10.1007/s12020-022-03112-w). Unfortunately, I cannot accept a manuscript that includes only two ACC cell models, one of which is SW13. The authors must replace SW13 with another, more appropriate ACC cell model. By the way the results are not in line with those obtained in H295R.

Response: We thank the Reviewer for his/her comment and the suggestion to consider alternative ACC cell models. However, we would like to respectfully clarify a few points regarding the use of the SW13 cell line in our study. The referenced article (Sigala et al. 2022. DOI: 10.1007/s12020-022-03112-w) does not state that SW13 is no longer acceptable as an ACC model; rather, it includes a reference about the beneficial effect of mitotane combined with radiotherapy on both H295R and SW13 adrenocortical cell lines (Ref. 54; Cerquetti et al.), indicating the relevance in some experimental contexts.

Indeed, compared to H295R cells SW13 did not produce adrenocortical hormones, but in line with real life, this can be observed in human tumours as well.

Also, based on Pubmed literature, SW13 continues to be used in current adrenocortical carcinoma research, particularly in pharmacological studies. In fact, even in recent years (2022–2025), several peer-reviewed, PubMed-indexed publications have used SW13 cells in ACC research, either alone or alongside H295R, such as:

  • Li et al., 2025, Cancer Biomark (DOI: 10.1177/18758592241308440)
  • Zou et al., 2024, Clin Transl Med (DOI: 10.1002/ctm2.70182)
  • Lai et al., 2024, Discov Med (DOI: 10.24976/Discov.Med.202436190.203)
  • Situ et al., 2024, Cancer Biol Ther (DOI: 10.1080/15384047.2024.2428469)
  • Vitalini et al., 2024, Nat Prod Res (DOI: 10.1080/14786419.2023.2258437)
  • Nocito et al., 2023, Cancers (DOI: 10.3390/cancers15041050)
  • Laha et al., 2022, J Clin Exp Cancer Res (DOI: 10.1186/s13046-022-02464-5)

In our study, we utilized experimental data of three cell models: H295R, SW13, and HAC15. While the reviewer is correct that results varied between cell lines, this difference was not major and it is expected highlighting the biological heterogeneity characteristic of ACC. Despite these differences, we observed that the combination of mitotane and defactinib was effective in both H295R and SW13 models.

Additionally, our study included a comprehensive series of experiments: two independent xenograft models, 2D monolayer cultures, 3D spheroids in ECM, 3D bioprinted tissue models, and transcriptomic analyses of both mitotane-sensitive and -resistant HAC15 cells, supplemented by human ACC tissue data. Given this extensive experimental framework, repeating all in vitro and in vivo experiments with additional cell lines would unfortunately exceed the logistical feasibility of our current study.

Nonetheless, we fully acknowledge the importance of validating findings in multiple relevant models and we explicitly address this as a limitation in the revised manuscript (please see on lines 312-328), while also emphasizing the robustness and multi-level nature of our current data.

Regarding the concentrations of mitotane and defactinib used in the 2D and 3D experiments, the authors must explain how these concentrations were selected. Were preliminary experiments conducted using increasing concentrations of the drugs, particularly defactinib? I recommend including the concentration–response curves, at least those obtained in monolayer cultures.

Response: We thank the Reviewer for their thoughtful comment and suggestion to include concentration-response data in the manuscript.

Regarding defactinib, to our knowledge, this compound has not previously been tested on adrenocortical carcinoma cell lines. Therefore, we conducted a literature review and found that defactinib was used at concentrations ranging from 0.1 to 20 µM in various cancer cell models, as reported by Le Large et al. 2021 (DOI: 10.1186/s13046-021-01892-z), Zhong et al. 2020 (DOI: 10.1007/s10495-020-01612-6), Kanteti et al. 2018 (DOI: 10.1080/15384047.2017.1416937), Lin et al. 2018 (DOI: 10.1002/pros.23476), and François et al. 2015 (DOI: 10.1093/jnci/djv123). Based on this, we conducted a viability assay using defactinib at 0.1, 0.5, 1, 2.5, and 5 µM concentrations in vitro. These concentration-response data have now been included in Supplementary Figure 4A.

Regarding mitotane, our group has previously studied its effects on adrenocortical cell viability, proliferation, and gene expression using a range of concentrations (10⁻⁴, 10⁻⁵, 5×10⁻⁶, and 10⁻⁶ M) as published by Zsippai et al. 2012 (DOI: 10.2217/pgs.12.116). In the literature, 5 µM mitotane is a widely used concentration in vitro for adrenocortical cancer cell lines, including in combination studies (e.g., Creemers et al. 2015 (DOI: 10.1210/jc.2016-2768). Schiavon et al. 2025 (DOI: 10.1016/j.biopha.2025.117917) also reported that most transcripts were reduced by half at 5 µM. Furthermore, this concentration has been applied in 3D culture models as well (e.g., Lindhe et al. 2010, DOI: 10.1055/s-0030-1261923). Accordingly, we selected 5 µM mitotane for our experiments.

Nevertheless, to address the reviewer’s suggestion, we have included comparative cell viability data for 5 µM and 15 µM mitotane versus DMSO control, which can be found in Supplementary Figure 1B of the revised manuscript. Also addressing the selection of concentration has been added to the revised manuscript as well, please found highlighted in the Results and Discussion section (lines 185-187 and 272-280).

In the in vivo experiments, there appear to be no significant differences between the group treated with defactinib alone and the group treated with the combination with mitotane. How do your results support the rationale for combining these treatments? Furthermore, from a clinical perspective, defactinib is metabolized by hepatic CYP3A4. Although mitotane is also primarily metabolized and cleared by the liver, one of its most notable effects is the induction of CYP3A4 activity. Therefore, mitotane could significantly influence drug metabolism and potential drug–drug interactions.

Response: We thank the Reviewer for his/her insightful comments. We agree that in our in vivo experiments, there were no statistically significant differences in tumour volume or number of macrometastases between the defactinib monotherapy group and the combination therapy group (defactinib + mitotane). However, both the defactinib monotherapy and the combination therapy groups showed significant improvements compared to the mitotane-only group, which more closely reflects the clinical context.

Mitotane and defactinib exert their effects through distinct molecular pathways - mitotane primarily inhibits steroid biosynthesis, while defactinib targets cell migration via FAK inhibition. Therefore, their combination offers complementary mechanisms of action. These findings suggest that adding defactinib to mitotane may enhance therapeutic efficacy compared to mitotane alone. We have emphasized this point more clearly in the revised Discussion section.

We also appreciate the Reviewer’s clinically important observation regarding drug metabolism. Indeed, mitotane is a known strong inducer of hepatic CYP3A4, while defactinib is metabolized via CYP3A4 and CYP2C9 [49]. As such, it is plausible that mitotane could accelerate the metabolism of defactinib, potentially reducing its plasma levels and therapeutic efficacy. Shimizu et al in a defactinib phase 1 study highlighted that preclinical evidence indicates that VS-6063 has a low potential for CYP3A-mediated drug interactions, in their study potent CYP3A4 inhibitors or inducers and systemic anticoagulation were prohibited (Shimizu et al. DOI: 10.1007/s00280-016-3010-1). In our study, no direct drug–drug interaction was observed in vitro using ¹H-NMR spectroscopy, and during our in vivo toxicity study of the combined treatment for three weeks did not showed clinical evidence of potential toxicity in mice. Nevertheless, in our in vivo model, we still observed a significant benefit of the combination therapy over mitotane monotherapy in terms of both tumour volume and metastatic burden.

Still, the possibility of a pharmacokinetic interaction in vivo in human underscores the importance of evaluating drug metabolism and interactions in future preclinical studies. Our current work focused on assessing therapeutic efficacy; however, we fully recognize the need for detailed pharmacokinetic investigations to better understand the impact of CYP3A4 induction on defactinib when co-administered with mitotane.

We have now included a discussion of this limitation and its implications in detail in the revised manuscript (please find highlighted in the discussion section, lines 329-345), highlighting the need for further evaluation before advancing to clinical application.

Minor points:

In the abstract (line 29), the phrase “but its impact on outcomes is limited” in reference to mitotane should be reconsidered. I understand that the authors aim to convey that patient outcomes in ACC remain poor despite standard treatments. However, mitotane has been shown to provide clinically significant benefits, particularly when surgery is not feasible or when used as adjuvant therapy to prevent or delay disease recurrence. Mitotane remains the cornerstone of systemic treatment for ACC, both in the unresectable/metastatic setting and as adjuvant therapy.

Response: We thank the Reviewer for this important comment. We referred to the ESMO-ENSAT guideline (Fassnacht et al. 2020) in the abstract, which state:

“Mitotane has been the reference drug for the management of ACC for decades and is increasingly used also in adjuvant settings following surgical removal of ACC. However, the value of this approach remains a matter of controversy because only a few studies have compared sufficiently large cohorts of treated and control patients.”

In agreement with the Reviewer’s suggestion, we have revised the highlighted sentence in the abstract to the following (please find highlighted):

“Mitotane, the only FDA-approved treatment for ACC, targets adrenocortical cells and reduces cortisol levels; and although it remains the cornerstone of systemic therapy, its overall impact on long-term outcomes is still a matter of ongoing clinical debate.”

In the introduction (line 66), the phrase “The use of cytotoxic drugs in ACC is not well-established [1]” is not clear to me. What do the authors mean by this sentence?

Response: We thank the Reviewer for this comment and the opportunity to clarify. The sentence in question - “The use of cytotoxic drugs in ACC is not well-established” - was based on the ESMO-ENSAT guidelines (Fassnacht et al., 2020), which state:
“The adjuvant use of cytotoxic drugs is not well established in ACC. Nevertheless, some centres are beginning to apply cytotoxic drugs (e.g. cisplatin plus etoposide) in selected patients with very high risk of recurrence and this approach is being investigated in a randomised trial.”

In this context, our statement refers specifically to the adjuvant (post-surgical) use of cytotoxic chemotherapy. While mitotane remains the primary systemic therapy and the use of cytotoxic agents (such as etoposide, doxorubicin, cisplatin) in advanced or metastatic ACC is more common, their application in the adjuvant setting is still considered investigational and not standardized across clinical practice.

To improve clarity, we have revised the sentence in the manuscript to:

„The clinical benefit of adjuvant use of cytotoxic chemotherapy in ACC is currently being explored in selected high-risk patients and clinical trials [1].”

Please find highlighted on lines 67-69.

Please improve the section of the introduction related to defactinib mechanism of action and its involvement in clinical trials.

Response: As requested by the Reviewer, we have revised and expanded the Introduction section to better describe the mechanism of action of defactinib and its clinical development, please find highlighted on lines 78-96:

Defactinib, a focal adhesion kinase (FAK) inhibitor, is currently investigated in several clinical trials: multiple phase I and II trials for advanced solid tumors, including KRAS-mutant non–small cell lung cancer and malignant mesothelioma [8,9]. It has been proven beneficial in high-grade ovarian and pancreatic cancer in phase 2 studies and is currently tested in phase 3 (NCT06072781). Single-agent FAK inhibitors have shown lim-ited efficacy, likely due to FAK's function as a signalling hub enabling cancer cell adapta-tion to treatment stress. However, combining FAK inhibitors with other therapies may be an effective therapeutic strategy [8]. Indeed, it is under investigation in combination regimens—for instance, with the RAF/MEK inhibitor avutometinib in recurrent low-grade serous ovarian cancer. Encouraging phase I data from this combination led to a Breakthrough Therapy Designation and initiation of a phase II trial [10]. Furthermore, defactinib is being studied in combination with immunotherapies, such as pembrolizumab, in patients with ad-vanced solid malignancies [11].

Defactinib (also known as VS-6063 or PF-04554878) is an ATP-competitive inhibitor of focal adhesion kinase (FAK) [9]. It functions by blocking FAK autophosphorylation - particularly at tyrosine residues Y397 and Y925 - thereby disrupting integrin-mediated signalling cascades, including the RAS/MEK/ERK and PI3K/Akt pathways, which are in-volved in tumour cell proliferation, survival, migration, and angiogenesis [12]. Preclinical studies have demonstrated potent anti-tumour activity of defactinib both in vitro and in vivo [12].”

Moreover, consider adding a sentence that connects its mechanism of action to the Wnt/β-catenin signaling pathway, preferably between lines 82 and 83. Alternatively, you may rephrase the sentences following line 83 to better establish this link.

Response: We have strengthened the potential link between FAK and Wnt signaling as follows (please find highlighted in the revised manuscript, lines 97-105):

„The Wnt/β-catenin signaling pathway plays a central role in ACC tumorigenesis, with activating mutations frequently observed [9], yet identifying effective Wnt inhibitors for clinical use remains a challenge [10,11]. Inhibiting aberrant Wnt/β-catenin activity disrupts tumor microenvironmental reprogramming and reduces ACC growth [11]. The interaction between FAK and Wnt signaling is highly context-dependent: in intestinal tu-morigenesis, FAK promotes Wnt activity, while in homeostasis, Wnt regulates FAK [10]. In different cancers, FAK can either modulate or act downstream of Wnt, and in mesotheli-oma, the two pathways are antagonistic [10]. These diverse interactions support the po-tential efficacy of FAK inhibition by defactinib in ACC. [10]”

In the results section (lines 168-169) the authors affirm “the combination treatment reduced spheroid size compared to control and mitotane-only groups, with an increased number of spheroids suggesting potential adhesion inhibition”. The increased number of spheroids should be reconsidered and replaced with something like “marked disruption of spheroid, reduction in compactness indicative of compromised cellular adesion”.

Response: We thank the Reviewer for this valuable observation. We agree that the original wording may have been misleading. Our intention was to describe the morphological changes observed under combination treatment, which included a loss of spheroid integrity and reduced compactness - features indicative of compromised cell-cell adhesion. Accordingly, we have revised the sentence in the Results section (lines 191-194) as follows:

“The combination treatment reduced spheroid size compared to control and mitotane-only groups, with marked disruption of spheroid structure and reduced compactness, suggesting compromised cellular adhesion.”

This revision better reflects the observed phenotype and aligns with the Reviewer’s recommendation. The text has been updated in the manuscript accordingly.

The Discussion section could be expanded and further developed. Additionally, the sentence in line 254 is identical to the one in the Introduction (line 81); consider rephrasing or removing it to avoid repetition.

Response: We have extended the Discussion section in accordance with the Reviewer’s suggestion, please find highlighted the changes. Additionally, we have rephrased the relevant part of the Introduction as requested, so the referenced sentence is no longer repeated in the revised manuscript. Please find highlighted all the changes.

Reviewer 3 Report

Comments and Suggestions for Authors

The manuscript of A. Patócs et al. is an interesting in vitro study that paves the way for repurposing of defactinib, a focal adhesion kinase inhibitor in order to treat a rare cancer with poor prognosis. The authors assembled 2D/3D in vitro systems to test the effect of the drug, consolidated by in silico analysis, and translated into an animal model. This comprehensive approach gives reliability to all conclusions of the study. Based on transciptomics and functional evaluations, it was revealed the prognostic significance of FAK signaling in the adrenocortical carcinoma models, and as well, it was highlighted the combined effect of two drugs: defactinib and mitotane.    

I strongly recommend publishing the paper, with some minor revisions:

Chapter 3. Discussions should be extended with additional literature data on FAK inhibitors repurposing (especially the available clinical data) and ACC treatment

Correlations between the experimental results derived from in vitro measurement, human samples and spheroid growth should be highlighted (if any).

Chapter 4.3 and 4.4. need some more experimental details, for reproducibility.

Author Response

REVIEWER 3

First, we would like to thank the Reviewer for his/her accurate work during the review process.

The manuscript of A. Patócs et al. is an interesting in vitro study that paves the way for repurposing of defactinib, a focal adhesion kinase inhibitor in order to treat a rare cancer with poor prognosis. The authors assembled 2D/3D in vitro systems to test the effect of the drug, consolidated by in silico analysis, and translated into an animal model. This comprehensive approach gives reliability to all conclusions of the study. Based on transciptomics and functional evaluations, it was revealed the prognostic significance of FAK signaling in the adrenocortical carcinoma models, and as well, it was highlighted the combined effect of two drugs: defactinib and mitotane.

I strongly recommend publishing the paper, with some minor revisions

Response: We grateful the Reviewer for his/her overall positive opinion on our work.

Chapter 3. Discussions should be extended with additional literature data on FAK inhibitors repurposing (especially the available clinical data) and ACC treatment

Response: In agreement of the Reviewer and the other Reviewer’s recommendation we extended the Discussion section regarding FAK inhibitors and clinical data. According the other Reviewer’s suggestion, some parts we included in the Introduction section about currently running clinical trials with defactinib (lines 78-90). However the whole disussion section have been reconstructed including data about defactinib treatment, combination therapies and trials on lung, ovarian, prostate and pancreatic cancer, and limitation of our study toembedd our findings more context regarding combined defactinib treatment in ACC. We hope that the novel version will meet the Reviewers requirements.

Correlations between the experimental results derived from in vitro measurement, human samples and spheroid growth should be highlighted (if any).

Response: We appreciate the Reviewer’s comment and would like to highlight the correlations observed across different experimental models regarding the combined therapy versus mitotane-only conditions. Specifically, combined treatment led to reduced cell migration in 2D monolayer cultures, impaired adhesion in 3D ECM-embedded spheroids, and decreased tumor size and number of macrometastases in the xenograft model. These findings are consistent with transcriptomic data from human ACC samples, where high FAK expression was associated with poorer progression-free and overall survival-supporting the role of FAK in promoting tumor progression through enhanced cell migration and metastatic potential.

We also included this into the Discussion section, please find highlighted on lines 284-291.

Chapter 4.3 and 4.4. need some more experimental details, for reproducibility.

Response: We have expanded the Materials and Methods section as requested. As the methodologies described in Sections 4.3 and 4.4 have already been applied in our previous publications, we have now included the additional experimental details in the Supplementary Materials and Methods to ensure full reproducibility. These additions are clearly highlighted in the revised manuscript.

We are grateful to the Reviewer for their thorough review, constructive feedback, and helpful suggestions. We hope that the revised manuscript now meets the requirements and expectations.

Round 2

Reviewer 2 Report

Comments and Suggestions for Authors

In their response letter, the authors implemented all the requested revisions and addressed the comments I had raised, though the adequacy of their responses varied. Although I had clearly stated my concerns regarding the use of SW13 cells as a model for ACC, the authors opted merely to cite previous studies in which these cells were employed. I understand that repeating all the experiments would be a substantial effort, and it is not my intention to impede the work of fellow researchers, but I sincerely hope that, in the future studies, the authors will more carefully evaluate the suitability of their cellular models and recognize that SW13 cells do not constitute an appropriate model for ACC.